# Current Understanding of the Role of Senescent Melanocytes in Skin Ageing

**DOI:** 10.3390/biomedicines10123111

**Published:** 2022-12-02

**Authors:** Bethany K. Hughes, Cleo L. Bishop

**Affiliations:** Barts and the London Faculty of Medicine and Dentistry, Blizard Institute, Queen Mary University of London, 4, Newark Street, London E1 2AT, UK

**Keywords:** melanocyte, senescence, skin, ageing, UV

## Abstract

Melanocytes reside within the basal epidermis of human skin, and function to protect the skin from ultraviolet light through the production of melanin. Prolonged exposure of the skin to UV light can induce irreparable DNA damage and drive cells into senescence, a sustained cell cycle arrest that prevents the propagation of this damage. Senescent cells can also be detrimental and contribute to skin ageing phenotypes through their senescence-associated secretory phenotype. Senescent cells can act in both an autocrine and paracrine manner to produce widespread tissue inflammation and skin ageing. Recently, melanocytes have been identified as the main senescent cell population within the epidermis and have been linked to a variety of skin ageing phenotypes, such as epidermal thinning and the presence of wrinkles. However, the literature surrounding melanocyte senescence is limited and tends to focus on the role of senescence in the prevention of melanoma. Therefore, this review aims to explore the current understanding of the contribution of senescent melanocytes to human skin ageing.

## 1. Introduction

### 1.1. Melanocyte Cell Biology

The term ‘melano-’ is derived from the Greek word ‘mélās’, which means black [1]. This denotes the ability of melanocytes to produce melanin, a pigment responsible for the colours of skin, eyes, and hair. Melanocytes are derived from neural crest cells and differentiate into melanoblasts, which move dorsolaterally to the ectoderm where they undergo differentiation into melanocytes [2]. Melanocytes are associated with a variety of pigmentation disorders, such as albinism, vitiligo, melasma, and senile lentigo, the effects of which can be seen prominently within the skin (reviewed by Yamaguchi et al., 2014 [3]). In the skin, melanocytes reside in the basal epidermis at a reported ratio of 1:5 to 1:10 with basal keratinocytes [4,5]. Melanocytes are highly dendritic, and thus can transfer their melanin to as many as 36 neighbouring keratinocytes [6]. Subsequently, melanin forms a supranuclear cap, surrounding the keratinocyte nucleus and protecting it from UV-induced DNA damage. This is particularly important for basal stem cells, to prevent both carcinogenesis and apoptosis, and to maintain a healthy stem cell pool for skin homeostasis [7].

Upon UV exposure, keratinocytes release alpha-melanocyte stimulating hormone (α-MSH), which acts through the G-protein coupled melanocortin-1 receptor present on the surface of melanocytes to stimulate the downstream transcription of microphthalmia-associated transcription factor (MITF) [8,9]. MITF is a transcription factor which stimulates the synthesis of tyrosinase, tyrosinase related protein 1 (TYRP1) and pre-melanosome protein PMEL (also known as PMEL-17, gp100, ME20, silver, HMB-45). These proteins are involved in the regulation of melanin synthesis, packaging, and transport, which has been extensively reviewed by Wasmeier et al., 2008 [8], and D’Alba et al., 2019 [10]. Interestingly, fibroblasts can also initiate melanogenesis after UV exposure, by upregulation of the cytokine growth differentiation factor 15 (GDF15), which in turn promotes β-catenin signalling, subsequent nuclear translocation of β-catenin, and upregulation of its downstream target MITF [11].

Melanin exists in two forms: brown/black eumelanin and yellow/red pheomelanin, and the ratio of these determines skin colour [10]. Both types of melanin are derived from tyrosine, which can be converted to L-DOPA and dopaquinone with the enzyme tyrosinase (Figure 1) [5]. Melanin is synthesised and packaged within melanosomes, which are early endosomal vesicles containing PMEL beta sheet scaffolds for melanin deposition, alongside tyrosinase and tyrosinase-related enzymes which are required for melanin synthesis (Figure 1) [8,10,12]. Mature melanosomes use microtubular transport to migrate to the distal portion of melanocyte dendrites. Here, the melanosomes can bind to cortical actin through Rab-27a/melanophillin/myosin VA complexing, to aid the transfer of melanosomes to neighbouring keratinocytes [8,13]. The mechanism of melanosome transfer to keratinocytes is unknown, but within the field four distinct transfer mechanisms have been proposed; cytophagocytosis, membrane fusion, shed vesicles, and exo/phagocytosis. This has been extensively reviewed in Moreiras et al., 2021 [14].

Melanocytes are intermittently mitotic cells, meaning they spend most of their life in a state of quiescence [15]. Upon signalling from neighbouring keratinocytes, such as the release of mitogenic factors α-MSH, endothelins 1 and 3, and basic fibroblast growth factor, melanocytes can resume the cell cycle [16,17]. In culture, mitogens, such as the phorbol ester phorbol-12-myristate-13-acetate (PMA), increase the proliferative capacity of melanocytes, allowing for a higher number of experimental cell stocks [18]. Interestingly, exosomes derived from keratinocytes are also capable of promoting melanocyte proliferation, tyrosinase activity, and melanin synthesis, while fibroblast-derived exosomes were unable to generate these consequences [19].

Melanocytes are perhaps best studied for their role in melanoma, described as the “deadliest form of skin cancer”, accounting for more than 80% of deaths from skin cancers [20,21]. Approximately half of melanomas exhibit a mutation in the serine/threonine kinase BRAF, most often a specific change of ‘T’ to ‘A’ nucleotide at codon 600 (V600E) [22]. Interestingly, mutations of BRAF V600E are commonly associated with the presence of benign melanocytic naevi or moles. These naevi have been reported to be positive for the senescence markers p16 and senescence associated beta galactosidase (SA-β-gal), suggesting senescence is a protective mechanism against melanoma development [23,24]. The links between melanoma and senescence have been extensively reviewed by Bennet, 2003 [25].

### 1.2. Senescence

Senescence was first reported by Leonard Hayflick in 1961, who found that human diploid fibroblasts in culture had a finite number of population doublings [26]. Subsequently, this has been attributed to telomere shortening with successive replications and has been termed replicative senescence (RS). Alongside RS, many other triggers can induce senescence, including oxidative stress, mitochondrial dysfunction, DNA damage caused by UV light exposure, and the activation of oncogenes (known as oncogene-induced senescence, or OIS) [27]. The presence of senescence markers is context dependent and there is no single marker of senescence. Commonly used markers include, but are not limited to cyclin-dependent kinase inhibitors p16 and p21, tumour suppressor protein p53, the presence of SA-β-gal, loss of the nuclear envelope protein Lamin B1, extracellular secretion of the Alarmin protein high mobility group box 1 (HMGB1), markers of DNA damage, including γH2AX and 53BP1, the presence of a senescence associated secretory phenotype (SASP), and morphological changes, such as an increased cell and nuclear area, and a decreased cell number due to proliferative arrest (measured by decreased Ki-67, or decreased incorporation of bromodeoxyuridine (BrdU) in DNA) [28,29,30,31]. This proliferative arrest prevents propagation of damaged cells, including inhibiting the formation of tumours. However, senescent cells are ‘double-edged’, as the development of a SASP can lead to paracrine senescence spread, widespread tissue inflammation and disease, and paradoxically can promote tumorigenesis [32]. Senescent cells accumulate with age and are linked to a variety of age-related diseases such as glaucoma, diabetes, pulmonary fibrosis, and cancers [33,34]. To combat this detrimental role for senescence, senolytic and senostatic drugs have been developed. The former selectively clear senescent cells, and this selective clearance has been found to improve both lifespan and healthspan in mice, with a reduction of age-related diseases such as sarcopenia, cataracts, and skin-related changes including increased dermal adipose tissue thickness, and improved coat condition. [35,36,37]. Senescent cells are also associated with human skin ageing phenotypes, for example epidermal thinning, wrinkles, melasma, and senile lentigo (colloquially known as age spots or liver spots) [11,38,39,40,41,42,43,44,45,46,47,48]. There is increasing evidence to suggest that senescent melanocytes play a significant role in aged skin. Therefore, this review will focus primarily on the current understanding of senescent melanocytes within human skin ageing.

## 2. Human Skin Ageing

Human skin has a myriad of roles, including protecting the body from pathogens, maintaining water content through the presence of lipids and filaggrins, controlling pH, and regulating body temperature through sweat glands [49]. The inner dermis consists of an extracellular matrix (ECM) rich with collagens (primarily collagen I and III), elastin, fibronectin and vitronectin (Figure 2) [50]. This ECM is responsible for aiding in tissue stability and repair and is commonly disrupted in aged skin. For example, solar elastosis is a condition described by accumulation of dysfunctional elastin fibres with increased levels of sun exposure, leading to accumulation of fine wrinkles [38]. Collagen synthesis, deposition, and turnover are also disrupted with age, including a reduction in pro-collagen I synthesis, a disordered collagen matrix with short, thin collagen fibrils, and increased matrix metalloproteinases (MMPs) for collagen degradation [51,52,53]. The dermis is home to papillary and reticular fibroblasts, with papillary fibroblasts believed to contribute more to skin ageing, perhaps due to their increased proliferative capacity [54].

The epidermis consists of five distinct layers of keratinocytes: the stratum basale, stratum spinosum, stratum granulosum, stratum lucidium, and stratum corneum [55]. Keratinocytes migrate along a calcium gradient until they reach the outer stratum corneum and become anucleated, terminally differentiated cells. Here, they aggregate with lipids to form a barrier to prevent excessive water loss [56]. Within aged skin, there is impaired terminal keratinocyte differentiation and reduced lipid formation, which can cause dry skin [57]. Although the basal stem cell population remains consistent with age, epidermal thinning is a key phenotype of skin ageing [39]. Grove et al., detected a faster loss of dansyl chloride dye from the stratum corneum of young volunteers compared to older volunteers, suggesting a decreased epidermal turnover and loss of keratinocyte proliferation with age [58]. Interestingly, there is also a loss of melanocytes with age, which could contribute to this epidermal thinning. However, the small number of melanocytes present in healthy human skin mean that this is unlikely. Gilchrest et al., reported a two-fold increase in melanocyte density in sun-exposed skin biopsies from donors, suggesting an intrinsic protective mechanism against UV. However, comparing the density from donors aged 28–80, the group discovered an age-related decline of around 6–8% in melanocyte numbers in both sun protected and sun exposed skin biopsies [59].

Skin ageing can occur through both intrinsic factors, such as genetic changes, replicative exhaustion, and hormonal disruption, and extrinsic factors. These include solar radiation, air pollution, tobacco smoking, poor nutrition, and miscellaneous (stress, lack of sleep, and high temperatures) [40]. Interestingly, intrinsic and extrinsic changes can lead to different phenotypes, and a combination of these determines overall skin ageing patterns. For example, intrinsic ageing leads to epidermal thinning, elastin loss, and a disrupted dermal epidermal junction (DEJ), whereas extrinsic ageing is characterised by deep wrinkles, solar lentigo (age spots), and thinning or thickening of the epidermis [42,43]. Experiments which aim to uncover differences between intrinsic and extrinsic ageing rely on the use of skin biopsies from either sun exposed or sun protected areas, as one of the main drivers of extrinsic skin ageing is exposure to UV light. This light exists in three wavelengths: UVA (320–400 nm), UVB (290–320 nm), and UVC (100–280 nm). However, UVC is unable to pass through the ozone layer due to its short wavelength [60]. Both UVA and UVB are considered phototoxic, because of their production of Reactive Oxygen Species (ROS) and the resulting oxidative stress. However, UVB is more potent and can directly induce DNA damage by the formation of cyclobutane pyrimidine dimers (CPDs) [60,61]. The role of UV in DNA damage has recently been reviewed by Chen et al., 2021, who discuss UV-induced ROS formation and its subsequent impact on skin ageing [62]. Furthermore, DNA damage is a common trigger for cellular senescence, particularly within sun exposed skin [27,43].

## 3. Senescence and Human Skin Ageing

Cellular senescence has been associated with organismal ageing in a variety of species, including *Drosophila*, rodents, primates, and humans [36,37,63,64,65,66,67]. Skin biopsies from older baboons showed an exponential increase in colocalised DNA damage markers, γH2AX and 53BP1, with increasing age, particularly within the telomeres (now known as telomere-associated foci or TAFs) [64]. An increased senescent cell burden has also been detected in rodents, with higher expression of the *Ink4a/Arf* locus (p16/Arf) detected in aged mice [68]. Seminal research from the Van Deursen group detected that selective clearance of p16-positive cells in aged rodents improved several age-related diseases, and increased rodent lifespan [36,37]. This work highlights the importance of senescence research to better understand and treat ageing across species. In humans, senescence was first detected within the skin, where skin biopsies from older donors had increased dermal SA-β-gal staining compared to younger donors, suggesting an increased senescent fibroblast burden with age [66]. Interestingly, p16-positive, non-proliferative senescent fibroblasts increase with age in human skin, suggesting that senescent triggers are present in the dermis [67]. Fibroblasts are undoubtedly the most thoroughly studied cell type within senescence research, due to their availability and relatively simple culture conditions. Senescent fibroblasts have also shown to contribute to a variety of skin ageing phenotypes through the secretion of a distinct, trigger-dependent SASP [41,69,70,71,72,73]. Alongside cell autonomous effects, the SASP can act in a paracrine manner to induce senescence in neighbouring cells [74,75]. Through this, senescent fibroblasts can trigger impaired keratinocyte barrier function and epidermal thinning, highlighting crosstalk between different cell types within the skin [75].

More recently, senescent cells have been discovered in the epidermis. The existence of senescent keratinocytes in vivo is controversial, due to their rapid turnover (reviewed in Low et al.) [48]. Sun-protected upper-inner arm skin was collected from subjects aged 46 to 81 years, who were offspring of nonagenarians from the Leiden Longevity Study [76,77]. Controls were taken from their partners, who were environmentally and age-matched. The nonagenarian’s offspring had lower p16-positive cells in both the dermis and the epidermis compared to their age-matched partners. Interestingly, the number of p16-positive epidermal cells was positively correlated with cardiovascular disease (CVD), but the number of p16-positive dermal fibroblasts was not. More recently, the group has identified that these p16-positive epidermal cells are melanocytes, and that the number of p16-positive melanocytes correlates with chronological and perceived age, while the number of p16-positive fibroblasts does not [78]. P16-positive melanocytes were also associated with changes in papillary dermal elastic fibres, with an increased number and length of the fibres [78]. This work has been recapitulated by Victorelli et al., who also found an increase in p16-positive melanocytes in the sun-protected epidermis with age [79]. Again, all p16-positive cells were also positive for the melanocyte markers S100 and Melan-A. This emerging evidence highlights the crucial need to understand the mechanisms of melanocyte senescence, and further explore their impact on ageing skin in vivo.

## 4. Melanocyte Senescence

### 4.1. Mechanisms

The mechanisms of melanocyte senescence have been extensively explored in the context of melanoma, particularly BRAF V600E OIS within naevi (moles) [22,25,80,81]. Although the focus of this review is on the contribution of senescent melanocytes to skin ageing, mechanistic insights can be gained from melanoma research. For example, knockdown of the known senescence biomarker *LMNB1*, a gene that encodes the nuclear lamina scaffold protein Lamin B1, induces senescence in human melanoma cells [82]. In addition, the Melanoma Inhibitory Activity (MIA) protein has been found to be expressed in BRAF V600E mutant melanocytes, suggesting a potential role for MIA in melanocyte OIS [83]. However, lentiviral transduction of human melanocytes with MIA did not induce senescence, suggesting MIA may be a downstream effector of senescence. This was confirmed with siRNA knockdown of MIA in senescent melanocytes, which decreased the number of SA-β-gal positive melanocytes, and decreased p21 mRNA expression, indicating that MIA is required for the maintenance of BRAF V600E OIS [83].

Perhaps most well explored in the context of melanocyte senescence is the role of the tumour suppressor protein p16^INK4a^ (p16). Encoded by the *CDKN2A* gene, which is situated on the short arm of chromosome 9, p16 acts by inhibiting cyclin dependent kinases 4 and/or 6, preventing phosphorylation of the retinoblastoma protein (Rb) and halting the cell cycle in G1 [84]. Mutations of *CDKN2A* (p16) are an established cause of melanoma, with Bartkova et al., reporting mutations in the p16/Rb pathway in all 22 human melanoma cell lines tested. This suggests that wild type p16 may act as a melanoma tumour suppressor through the induction of senescence. Gray-Schopfer et al., investigated the role of p16 in senescence of three normal human melanocyte primary cell cultures. Using hTERT (telomerase reverse transcriptase) retrovirus to prevent telomere shortening, the group found that melanocyte immortalisation could only take place with an additional disruption to p16, again suggesting p16 is able to defend melanocytes from melanoma. This was further evidenced by the presence of p16 in all naevi sampled by the group, which were also positive for other senescent markers such as SA-β-gal, and the presence of senescence-associated heterochromatin foci (SAHF). Interestingly, immunofluorescent (IF) staining revealed that only one naevus presented low levels of p53, and no naevi stained positively for p21. This suggests that the mechanism of senescence in naevi was mediated by the p16/Rb pathway but not the p21/p53 pathway.

Intriguingly, p16-deficient melanocytes can engage senescence via p14^Arf^ (Arf), a protein also encoded on the *CDKN2A* gene via an alternative reading frame [85]. Arf is an established senescence inducer, through sequestering p53-inhibitor mouse double minute 2 (MDM2). Subsequently, p53 can act as a transcription factor for p21^cip1^ (p21), which inhibits cyclin-dependent kinase 2 and Rb phosphorylation, inhibiting cell cycle progression [86]. Ha et al., used a cutaneous malignant melanoma mouse model, Hepatocyte Growth Factor/Scatter Factor (HGF/SF) mice, and cross bred these mice with a deficiency for Arf (*Arf^−/−^*), p16 (*Ink4a^−/−^*), or both [87]. The mean onset age of melanoma from *Ink4a^−/−^* or *Arf^−/−^* mice was significantly lower than their wildtype counterparts, suggesting that both proteins play a role in protecting against melanoma. Losing only one *Arf* allele (*Arf^+/−^*) did not significantly alter melanoma incidence, but *Ink4a^+/−^* did induce a younger onset. The group also cross-bred the mice with *TP53^−/−^* mice, but these mice died of other cancers before succumbing to any melanoma. To overcome this, the group generated an HGF/SF to C57BL/6 crossover, allowing mice deficient in p53 to live longer to fully assess the incidences of melanoma. Interestingly, after 129 days, no melanoma developed in the *TP53^−/−^* mouse. The group hypothesised that Arf is acting independently of p53 as a potent tumour suppressor and senescence inducer in these mice. This was confirmed when melanocytes isolated from *Arf^−/−^* mice did not reach senescence in culture over three months, whereas melanocytes isolated from wild type mice underwent replicative senescence after four weeks. The growth of *Ink4a-Arf*^−/−^ cells with a subsequent p53 knockdown was not significantly different from that of *Ink4a-ArF^−/−^* cells with intact p53, again highlighting that p53 does not play a significant role in replicative melanocyte senescence or growth.

Conversely, other groups have found that p16-deficient melanocytes are capable of engaging senescence through the p21/p53 pathway. Sviderskaya et al., isolated melanocytes from two p16-null patients, which grew in vitro for approximately 44 population doublings [88]. This is 34 more population doublings than wild type adult melanocytes cultured previously in the same medium, suggesting p16 induced an earlier cell cycle arrest in the wild type melanocytes. Upon reaching growth arrest, the p16 null melanocytes expressed increased p21 and p53 proteins, suggesting delayed engagement of the p21/p53 pathway to induce senescence. Elevated levels of p21 and p53 were only observed in melanocytes which were lacking p16, and not in wild type cells, perhaps due to temporal differences in engagement of the pathways [88,89]. This work suggests that p16 is the primary driver of replicative melanocyte senescence, but that p53/p21 can initiate a delayed response. More recently, it has been suggested that p21 could induce melanocyte senescence independently of p53. For example, H_2_O_2_ treatment of primary human foreskin melanocytes induced an upregulation of p21, but no change in total cell protein levels of p16 or p53, as seen by Western blotting and IF [90]. When p21 was knocked down in the melanocytes by siRNA, the number of senescent cells was reduced, quantified by a reduction in SA-β-gal, and an increase in proliferation assessed by EdU incorporation into newly synthesised DNA. Using RNA sequencing of these senescent melanocytes, the group found 750 differentially expressed genes (DEGs) between H_2_O_2_-treated and control cells [90]. With Kyoto Encyclopedia of Genes and Genomes (KEGG) pathway analysis, Hou et al., identified mitogen activated protein kinase (MAPK) pathways as differentially regulated within the senescent melanocytes. Upon validation, the group discovered an increase in ROS-mediated phosphorylation of ERK1/2 map kinases, which are involved in the cell cycle transition from G1 to S phase. Dendrite formation and adhesion was also impaired in H_2_O_2_-treated cells, suggesting they may exhibit impaired melanosome transfer [90].

Interestingly, melanin is considered to be a main driver of melanocyte senescence, perhaps due to the pro-oxidative nature of melanin formation [91]. Jenkins and Grossman investigated this by inhibiting the oxidation of L-tyrosine to dopaquinone, the first step in melanin synthesis (Figure 1). Inhibition with phenylthiourea (PTU) caused a reduction in ROS levels in both wild type melanocytes and in melanocytes with p16 depletion by siRNA, despite their increased basal ROS. When exposed to UV, melanin itself is able to cause single strand breaks and ROS within melanocytes, which could lead to senescence induction through oxidative stress and DNA damage [92]. This is supported by San Juan et al., who found that lentiviral overexpression of MYC protooncogene in neonatal melanocytes induced an initial melanocyte differentiation, and subsequent senescence after 15 days [93]. This process correlated with increased levels of intracellular melanin, and increased expression of the DNA damage marker γH2AX. Furthermore, senescent melanocytes contain increased amounts of melanin, and the activation of alpha-melanocyte stimulating hormone (which initiates cAMP controlled melanogenesis) causes senescence induction in both Black and Caucasian skin [94]. Melanocyte senescence has recently been investigated as a driver of pigmentation disorders (reviewed in Kim et al., 2022) [47]. However, the role of senescent melanocytes themselves in pigmentary disorders appears to be limited, with melanin induction occurring in a paracrine manner, particularly from senescent fibroblasts which are known to induce senile lentigo and melasma [44,45,46].

### 4.2. UV-Induced Melanocyte Senescence

The protective role of melanocytes against UV light is somewhat counteracted by UV-induced melanocyte senescence. However, melanocytes are more resistant to UV rays, generally requiring a higher dose than fibroblasts to induce a stable senescence response [95]. Unsurprisingly, the mechanism of UV-induced melanocyte senescence is unique compared to both replicative senescence and OIS. Barker et al., discovered that after human melanocytes were irradiated with a single dose of UVB, there was an increase in p53 expression after 4 h [96]. Interestingly, in lightly pigmented melanocytes, p53 expression continued to increase up to 48 h after UV irradiation, but heavily pigmented melanocytes began to lose their p53 expression after 24 h. The lightly pigmented melanocytes also contained more CPDs, highlighting that lightly pigmented melanocytes are more susceptible to damage. Subsequently, the group detected a sustained increase in p21 expression over 48 h, and a decrease in phosphorylated Rb, but did not investigate whether there were changes in p16 expression [97].

The role of p53 in UV-induced melanocyte senescence has also been recapitulated by other groups. For example, Choi et al., investigated the effects of two repeated doses of UVB on neonatal human melanocytes, detecting induction of senescence by measuring an increased area of cell bodies, decreased cell proliferation, and increased SA-β-gal activity [98]. Interestingly, UVB-induced senescent melanocytes also exhibited a higher melanin content after maintaining the cells for four weeks, which was not seen in late-passage melanocytes. Through droplet digital PCR, the group found an increased abundance of *CDKN1A* (p21) and *TP53* in the UVB treated cells compared to controls, again suggesting that this trigger induces senescence by the p53/p21 pathway. This was also seen at a protein level via immunoblotting, with p53 increasing up to 48 h and then decreasing, although not returning to control levels [98]. Intriguingly, p16 was also increased at the protein level 2 weeks post-UVB irradiation, perhaps suggesting a role for p16 in maintaining the senescent phenotype. However, p16 levels were not reported at any earlier timepoints. Inhibiting p53 with pifithrin-α (PFTα) two weeks after UVB exposure reduced both tyrosinase and melanin content, suggesting a potential role for p53 in pigmentation. Murase et al., have also reported the involvement of p53 in hyperpigmentation, discovering an increase in *TP53* mRNA by RT-PCR in hyperpigmented skin spots from epidermal punch biopsies [99].

Zhang et al., discovered an earlier role for p16 in UV-induced melanocyte senescence, detecting an increase in p16 protein six hours after a single dose of UVB [100]. The group also found an increase in p21 and p53, and a higher expression of SASP factors, such as IL-6 and IL-8, as well as an increased number of γH2AX foci, increased SA-β-gal activity, and an increased number of CPDs. When melanocytes were treated with UVB twice per week for two weeks, the expression of p53, p21, and p16 remained high. However, these were lost during the subsequent two weeks without UVB exposure, despite the cells maintaining a significantly higher expression of SA-β-gal at this time. Intriguingly, upregulation of interferon stimulatory genes, including MX Dynamin like GTPase 1 (MX1) and signal transducer and activator of transcription 1 (STAT1), was sustained at this 28-day timepoint in lightly pigmented melanocytes, but lost in darkly pigmented melanocytes, potentially providing additional protection to the lightly pigmented melanocytes against further UV insults. Interferon gamma signalling is known to induce the extracellular release of the senescence marker HMGB1, a high mobility group protein which is involved in chromatin remodelling. HMGB1 subsequently binds to receptor for advanced glycation endproducts (RAGE), which can trigger further inflammation through the NFκB pathway [101]. Zhang et al., discovered that melanocytes upregulate RAGE expression three hours after a single UVB dose and upregulate HMGB1 after 24 h [100]. The extracellular secretion of HMGB1 may act in both an autocrine and a paracrine manner to deepen senescence within the tissue and protect the cells from propagating damaged DNA. Other groups have discovered a role for HMGB1 following different triggers of senescence associated with DNA damage. For example, H_2_O_2_-treated melanocytes showed nuclear to cytoplasmic translocation and subsequent extracellular release of HMGB1 [102]. HMGB1 can also be released from keratinocytes upon UV stimulation, acting as a DNA damage ‘warning signal’, and upregulating melanosome transfer and melanocyte dendricity [103].

Although informative, a single UV dose does not recapitulate melanocyte UV exposure in vivo. For this reason, Martic et al., treated melanocytes twice daily for four consecutive days with 0.125 J/cm^2^ of UVB. The group were able to induce senescence, and observed the characteristic increased cell size, cessation of the cell cycle, increased levels of p53 and p21, increased melanogenesis, and increased SA-β-gal staining. The group also investigated the presence of Lamin B1, and found that following a transient increase, the levels of nuclear Lamin B1 were downregulated following UVB treatment. Furthermore, proteasome impairment was detected using a GFP-degron expressing reporter cell line which should be degraded with a functional proteasome. This highlighted that UVB-induced senescent melanocytes exhibit another hallmark of cellular ageing [104]. Martic et al., also investigated the paracrine impact of UV-induced senescent fibroblasts on melanocytes, by using a three-day transwell co-culture system [95]. The group discovered that melanocytes cultured with senescent fibroblasts had increased melanogenesis but did not see any change in melanocyte proliferation or cell area. The group did not report any investigations into the expression of p16. The paracrine effect of UVB-induced senescence has also been investigated in other contexts. For example, Victorelli et al., explored the paracrine effect of UVA + UVB senescent melanocytes on neighbouring keratinocytes, using a ‘melanoderm’, an LSE containing melanocytes and stratified keratinocytes [79]. Keratinocytes grown in melanoderms with UV-induced senescent melanocytes exhibited a greater number of TAFs, an established marker of DNA damage-induced senescence [105]. The keratinocytes also had reduced proliferation (measured by decreased Ki-67), an increased p16 expression, and the melanoderms had a reduced epidermal thickness. Investigation of paracrine cross-talk within skin cell populations in the epidermis provides important insight into the role of senescent melanocytes in vivo.

### 4.3. Evidence That Senescent Melanocytes Impact Human Skin Ageing In Vivo

To date, there are very few publications which link melanocyte senescence to human skin ageing in vivo. However, seminal research within the Leiden Longevity Study showed that melanocytes were the main epidermal senescent cell population in vivo (discussed within Section 3) [76,77,78,79]. In 2019, Victorelli et al., further explored the impact of senescent melanocytes within human skin [79]. Alongside finding an increased number of p16-positive melanocytes in older skin biopsies, the group discovered a significant decrease of HMGB1 expression with age, but no change in expression of p21. This is perhaps unsurprising, as these skin biopsies were taken from sun-protected areas and are not exposed to UV. When investigating DNA damage, the group saw an increase in the number of γH2AX foci in melanocytes with age, and importantly an increase in TAFs, suggesting telomere-specific damage. Using telomere Q-fish, the group were able to identify that the telomeres within older skin biopsies are not significantly shorter than their younger counterparts. Furthermore, slightly shorter telomeres did not correlate with an increasing number of γH2AX foci. This suggests that telomere shortening may not be a key driver of melanocyte senescence in vivo.

Following this work, Victorelli et al., investigated whether senescent melanocytes could have a paracrine effect on neighbouring cells, contributing to the overall phenotypic changes of skin ageing [79]. They discovered that within human skin biopsies, the keratinocytes surrounding melanocytes with TAFs had more TAFs themselves, suggesting that the senescent melanocytes were able to induce a secondary DNA damage response in nearby cells. This was also recapitulated within melanoderms, in vitro models containing UVA + B induced senescent melanocytes and stratified keratinocytes. Melanoderms containing senescent melanocytes also had fewer keratinocytes, and a decreased expression of the proliferation marker Ki-67 in keratinocytes (see Section 4.2). Consequently, melanoderms with senescent melanocytes had a thinner epidermis, a key phenotype of skin ageing. This suggests that senescent melanocytes may be fundamental drivers of skin ageing.

To uncover the mechanism of melanocyte-driven paracrine senescence, the group used a cytokine array to investigate the SASP of senescent melanocytes and found a significant increase in the secretion of interferon gamma-induced protein 10 (IP-10) [79]. IP-10 is a chemokine which is involved in the immune response, by binding to its receptor CXCR3. Interestingly, the group found an increase in the expression of CXCR3 by IF in the basal epidermis of older donor skin. As a proof of concept, the group added SASP conditioned medium from senescent melanocytes to human dermal fibroblasts (HDFs) in culture and found a three-fold increase in fibroblast TAFs. When these HDFs were treated with AMG487, a direct inhibitor of CXCR3, the number of TAFs was rescued to control levels. Directly adding recombinant IP-10 to HDFs also induced TAFs, and the group detected a significant increase in mitochondrial ROS using mitoSOX. This suggests that increased IP-10 secretion and increased binding to CXCR3 within neighbouring basal cells could induce a paracrine DNA damage response via increased ROS. Finally, Victorelli et al., wanted to investigate the therapeutic potential of senolytic drugs, which selectively kill senescent cells. Within the melanoderm, treatment with the Bcl-2 and Bcl-xL inhibitor ABT-737 cleared senescent melanocytes. Importantly, the addition of the senolytic rescued epidermal atrophy, and reduced the number of TAFs within keratinocytes. This provides substantial evidence that senescent melanocytes may significantly contribute to human skin ageing, and emphasises the requirement for further research to tackle skin ageing.

## 5. Conclusions

Due to the paucity of research surrounding melanocyte senescence and skin ageing, it is difficult to draw definitive conclusions about the role of melanocytes within aged skin. However, recent evidence suggests that melanocytes are the main senescent cell population within the epidermis of aged skin, highlighting an urgent need for research focus within this area. Although information may be drawn from oncogene-induced senescence in the context of melanoma, and that these senescent cells undoubtedly contribute to ageing phenotypes through their secretome, the heterogeneous nature of senescence means that this information may be of limited use when considering other triggers, such as DNA damage-induced, paracrine, or replicative melanocyte senescence. It is currently unclear as to which trigger is most prevalent in human skin with age, and the small number of melanocytes which exist in human skin provide limited statistical power to answer this question. Furthermore, a greater understanding of each trigger could provide novel biomarkers that could aid this understanding. In summary, there is strong emerging evidence that senescent melanocytes exist within aged skin in vivo and contribute to skin ageing phenotypes. Continued research will provide elucidation of the mechanisms involved and allow for tailored senotherapeutics to improve aged skin.

## Figures and Tables

**Figure 1 biomedicines-10-03111-f001:**
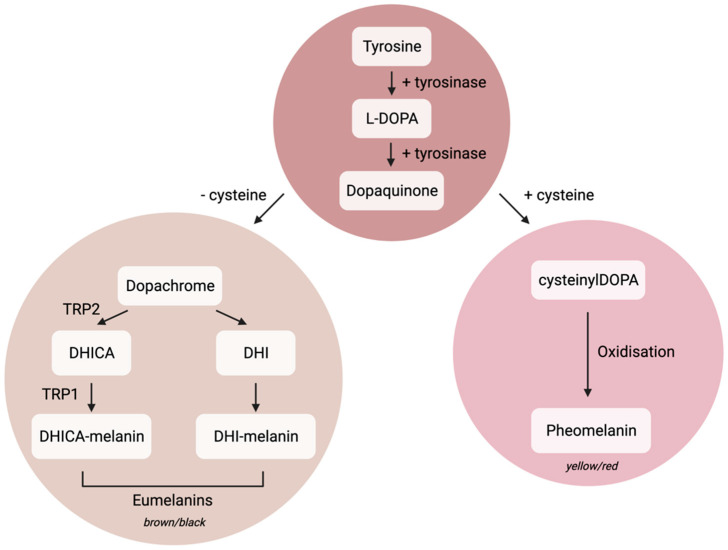
Eumelanin and pheomelanin production. Adapted from Cichorek et al., 2013 [5]. Tyrosine is converted to levodopa (L-DOPA) and dopaquinone by the enzyme tyrosinase. Here, the pathway diverges dependent on the presence of cysteine to form either brown/black eumelanins or yellow/red phenomelanins. During eumelanin production, dopachrome can be converted to 5,6-dihydroxyindole-2-carboxylic acid (DHICA) melanin, facilitated by the presence of tyrosinase-related proteins 1 and 2 (TRP1 and TRP2). To form pheomelanin, cysteinylDOPA is oxidised.

**Figure 2 biomedicines-10-03111-f002:**
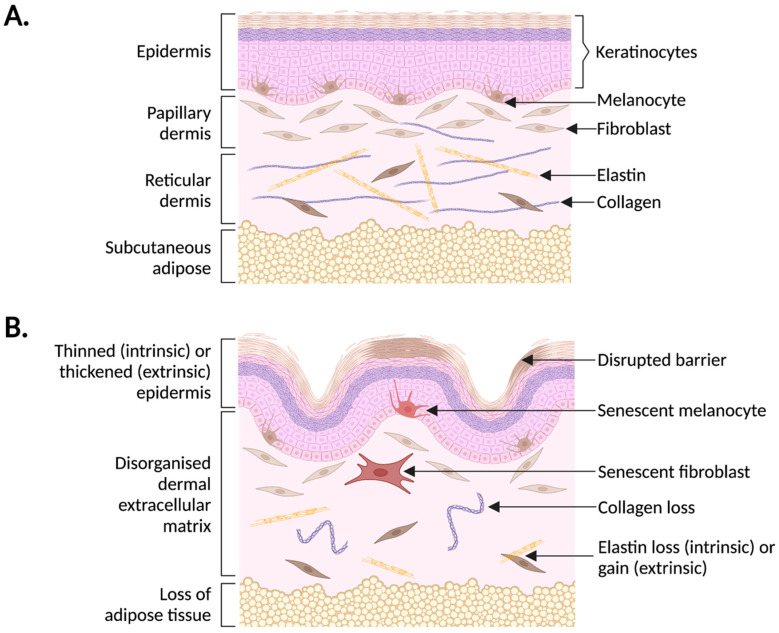
The structure of young versus old skin. (**A**) Human skin is divided into an epidermis, dermis, and subcutaneous adipose tissue. The epidermis contains stratified layers of keratinocytes, and melanocytes within the basal layer. The dermis is further divided into a fibroblast-rich papillary layer, and an ECM-rich reticular layer containing collagens and elastin. (**B**) With age, the epidermis may become thinner (intrinsic ageing) or thicker (extrinsic ageing), the stratum corneum barrier becomes dysfunctional, and there is the presence of senescent melanocytes within the basal layer. The dermal matrix becomes disorganised, with a loss of collagen and changes in elastin content, and an accumulation of senescent fibroblasts. There is also a loss of adipose tissue.

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
