# Peer review of "Current Understanding of the Role of Senescent Melanocytes in Skin Ageing"

_biomedicines, 2022, doi:10.3390/biomedicines10123111_

Round 1
Reviewer 1 Report
Review of manuscript ‘Current understanding of the role of senescent melanocytes in skin ageing’ by Bethany K Hughes and Cleo L Bishop sent to Biomedicines.
Comments
This is a review paper on the role of senescent melanocytes on skin ageing. After introducing melanocyte cell biology and senescence it gives a broad overview on possible currently known (path-) ways melanocyte senescence might influence skin cell ageing (in cellular skin models) and skin ageing in vivo. It is concluded that there is strong evidence that senescent melanocytes contribute to skin ageing phenotypes in vivo.
· The paper is well written and contains relevant references to the topic.
· It gives a good overview on the current status of knowledge without own thoughts, models or further own considerations.
Detailed comments, suggestions for minor changes
1. Page 5, line 160-162: The sentence ‘These include solar radiation, air pollution, tobacco smoking, nutrition, miscellaneous (stress, sleep, and temperature), and cosmetics [40].‘ is misleading. The extrinsic factor nutrition is not leading to skin aging, but the wrong nutrition is leading to eat. In the same manner: not stress or sleep or temperature per se leads to skin aging but too much stress, too low amount of sleep and too high temperatures. And, it is not the use of cosmetics that leads to skin aging. The use of cosmetics prevents the skin from aging as mentioned in the given reference [40].
2. Page 5, line 169-170: It is not the visible light that splits in UVA, UVB and UVC. Only the UV-part of the visible light splits into the three wavelengths ranges. And with UVB and UVC you do not really have light that is ‘visible’.
3. Page 5, line 175-176: A review mentioned by Chen et al. is mentioned. But, nothing is written about the content of Chen’s review. One or two sentence summary on the main content of this paper should be included.
4. Page 5, line 176-177: There is no reference given for the last sentence of this chapter.
Author Response
We thank the reviewer for taking the time to comment on our manuscript.
All the comments raised relate to text on page 5 on the manuscript. We have addressed each of these points in the revised manuscript.
Reviewer 2 Report
Please find attached file.

Author Response
We thank the reviewer for taking the time to consider our manuscript and for finding that it is ready for publication in its original format.